# Clinical Outcomes and Prolonged SARS-CoV-2 Viral Shedding in ICU Patients with Severe COVID-19 Infection and Nosocomial Bacterial Pneumonia: A Retrospective Cohort Study

**DOI:** 10.3390/jcm11226796

**Published:** 2022-11-17

**Authors:** Chuan-Yen Sun, Jia-Yih Feng, Jhong-Ru Huang, Hisao-Chin Shen, Yuh-Min Chen, Wei-Chih Chen, Kuang-Yao Yang

**Affiliations:** 1Department of Chest Medicine, Taipei Veterans General Hospital, Taipei 112, Taiwan; 2School of Medicine, National Yang Ming Chiao Tung University, Taipei 112304, Taiwan; 3Institute of Emergency and Critical Care Medicine, National Yang Ming Chiao Tung University, Taipei 112, Taiwan; 4Cancer Progression Research Center, National Yang Ming Chiao Tung University, Taipei 11221, Taiwan

**Keywords:** prolonged viral shedding, coronavirus disease 2019, secondary pneumonia, severe acute respiratory syndrome coronavirus 2

## Abstract

Objectives: This study explored the clinical outcomes and association of prolonged severe acute respiratory syndrome coronavirus 2 (SARS-CoV-2) shedding in patients with severe coronavirus disease 2019 (COVID-19) infection who developed nosocomial pneumonia. Methods: This was a retrospective study conducted in a medical center in Taiwan. From May to September 2021, patients from four intensive care units were enrolled after SARS-CoV-2 was confirmed through quantitative polymerase chain reaction and all cases were compatible with the definitions of severe COVID-19 infection. Baseline characteristics, disease severity, clinical outcomes, and times of viral shedding were recorded. Results: A total of 72 patients were diagnosed as having severe COVID-19 infection and 30 developed nosocomial pneumonia, comprising hospital-acquired pneumonia (HAP) and ventilator-associated pneumonia (VAP). The patients with severe COVID-19 infection and concomitant HAP/VAP had longer intensive care unit (ICU) stays and fewer ventilator-free days at Day 28. An independent risk factor for nosocomial pneumonia was a greater SOFA score at admission. Furthermore, the patients with severe COVID-19 infection who developed HAP/VAP had a significantly longer duration of SARS-CoV-2 shedding (19.50 days vs. 15.00 days, *p* = 0.006). Conclusions: Patients with severe COVID-19 infection who developed nosocomial pneumonia had longer SARS-CoV-2 shedding days, more complications, and worse outcomes.

## 1. Introduction

The coronavirus disease 2019 (COVID-19) has been declared a pandemic by the World Health Organization as of 11 March 2020. This disease is caused by severe acute respiratory syndrome coronavirus 2 (SARS-CoV-2) and has caused millions of deaths. Viruses, such as influenza and respiratory syncytial viruses, may damage ciliated cells, impair mucociliary clearance, and facilitate microorganism colonization in the airway [1,2]. Bacterial invasion occurs after airway epithelial cell death induced by viruses [2]. Viral and bacterial coinfection not only worsens clinical outcomes but also increases the length of ICU stay [3]. Hospital-acquired pneumonia (HAP) and ventilator-associated pneumonia (VAP) result in worse clinical outcomes and prolong the duration of ventilator use [4]. In ICU patients with VAP who have acute respiratory distress syndrome, the most frequently isolated pathogens from the airway were Gram-negative bacilli, including *Pseudomonas aeruginosa*, *Acinetobacter baumannii*, *Enterobacteriacae*, and *Stenotrophomonas maltophilia* [5]. In a multicenter study on critically ill patients with COVID-19 infection, 46% developed nosocomial infection, and Gram-negative bacteria accounted for 64% of bacteria isolated from patients with VAP [6]. Real-time reverse-transcriptase polymerase chain reaction (RT-PCR) is widely used to confirm SARS-CoV-2 infection through the detection of viral ribonucleic acid [7]. The cycle threshold, known as the Ct value, is the number of cycles required for RT-PCR products to reach a detectable level, with 40 generally considered the cutoff for a negative Ct value [8]. However, a positive PCR result does not indicate infection. Some studies regard 30 as the threshold for being non-infectious with no risk of transmission [9]. Recent studies have concluded that some factors are associated with prolonged viral shedding, including old age, female sex, and more days from symptom onset to admission [10]. To date, data on the association between SARS-CoV-2 prolonged shedding and viral–bacterial coinfection have been scant. This study explored the characteristics of patients who developed nosocomial bacterial pneumonia after contracting COVID-19 and the impact of viral shedding.

## 2. Methods

### 2.1. Study Design, Setting, and Patients

This retrospective observational study was performed in Taipei Veterans General Hospital (VGHTPE), a 2800-bed medical center in Taipei, Taiwan. Patients were enrolled from May to September 2021, corresponding to the period of a local outbreak of COVID-19 in Taiwan. During this period, critically ill patients with severe COVID-19 infection were admitted to four ICUs in VGHTPE, and only cases confirmed through SARS-CoV-2 RT-PCR were enrolled in the study. Severe COVID-19 infection was defined as dyspnea, respiratory rate > 30 breaths/min, blood oxygen saturation of 93% on ambient air, or a ratio of arterial oxygen partial pressure to the fraction of inspired oxygen of less than 300 [11]. Exclusion criteria were as follows: no RT-PCR confirmation of SARS-CoV-2, no ICU admission, age < 18 years, and concomitant human immunodeficiency virus (HIV) infection. The study protocol was approved by the Institutional Review Board of Taipei Veterans General Hospital (IRB number: 2021-12-006BC), and the requirement for informed consent was waived.

### 2.2. Data Collection and Measurements

Demographic data on age, sex, underlying diseases, laboratory results, and severity score at admission were collected. Information on treatment, namely the use of tocilizumab, remdesivir, and enoxaparin, was collected. Corticosteroid exposure was quantified as dexamethasone equivalents and measured by the cumulative dosage of dexamethasone in milligrams (mg) from admission to the first day of bacterial culture from airway specimens in the HAP/VAP cohort and from admission to ICU discharge or death in the non-HAP/VAP cohort. Clinical course and outcomes included the use of mechanical ventilation (MV), extracorporeal membrane oxygenation (ECMO), prone position and renal replacement therapy. Outcomes included gastrointestinal bleeding events, length of ICU stay, duration of SARS-CoV-2 viral shedding, ventilator-free days at Day 28, ICU mortality, and in-hospital mortality. In addition, sequential organ failure assessment (SOFA) scores [12] on the day of admission were recorded.

### 2.3. HAP and VAP Definitions

Bacterial pneumonia was defined using the IDSA/ATS 2016 HAP/VAP guidelines, which included radiographic evidence of new or progressive pulmonary infiltration, consolidation, or cavitation, as well as clinical evidence of respiratory symptoms, including unstable vital signs, new onset of purulent sputum, changes in sputum character, increased respiratory secretions, increased suctioning requirements, and worsening gas exchange [13]. Laboratory evidence comprised elevated white blood cell counts, C-reactive protein level, and procalcitonin level. Pathogens from respiratory specimens were collected by endotracheal aspiration or sputum induction and were cultured at least 48 h after hospitalization. HAP was defined as pneumonia occurring 2 days or more after admission. VAP was defined as pneumonia developing 2 days or more after mechanical ventilation [13]. Patients diagnosed as having HAP or VAP in this study all received at least a 5- to 7-day course of appropriate antibiotics confirmed by a detailed chart review.

### 2.4. SARS-CoV-2 RT-PCR

SARS-CoV-2 RT-PCR was performed using the Roche Cobas 6800 system (Roche Diagnostics, Rotkreuz, Switzerland). The selective amplification of target nucleic acids from samples was performed using target-specific forward and reverse primers for the ORF1ab non-structural region, which is individual to SARS-CoV-2. Another conserved region in the structural protein envelope E-gene was chosen for pan-Sarbecovirus detection. In this study, Ct values of both gene targets for all the RT-PCR tests of nasopharyngeal swabs, or endotracheal aspirates if patients were mechanically ventilated, were included as in a previous study [14]. Viral dynamics were calculated according to the Ct value of the SARS-CoV-2-specific target (ORF1ab). The result of RT-PCR was expressed as the Ct value, which was considered negative if ≥40. The limit of detection (LoD) determines the lowest detectable level of SARS-CoV-2 at which greater or equal to 95% of all replicates test positive. The LoD was determined via a cultured and isolated virus from a US patient (USA-WA1/2020, catalog number NR-52281, lot number 70,033,175, 2.8 × 10^5^ Median tissue culture infectious dose (TCID)_50_/mL). The concentration level with observed hit rates over or equal to 95% was 0.009 TCID_50_/mL for SARS-CoV-2 and the Probit-predicted 95% hit rate was 0.007 TCID_50_/mL (95% CI: 0.005–0.036). The concentration level in a dilution series with observed hit rates over or equal to 95% was 46 copies/mL for SARS-CoV-2. The Probit model 95% LoD was about 25 copies/mL (95% CI: 17–58 copies/mL).

The duration of SARS-CoV-2 shedding was defined as the number of days from symptom onset to the first day the Ct value reached >30 because some studies have regarded a value of 30 to be noninfectious, and no virus was cultured from samples when the Ct value was >30 [15,16]. Moreover, a Ct value of ≥30 is the benchmark for deisolation stipulated by the Taiwan Center for Disease Control [17]. Patients without a Ct value over 30 during their admission were right-censored at the time of their last Ct value.

Treatment outcomes were in-hospital mortality, length of ICU stay, ICU mortality, and ventilator-free days at Day 28. All the patients were followed up from admission to death or discharge, with the last patient discharged on 21 October 2021.

### 2.5. Statistical Analysis

Results are presented as medians with interquartile ranges (IQRs) or numbers with percentages wherever appropriate. The Mann–Whitney U test was used to compare non-normally distributed continuous variables. The Pearson χ^2^ test or Fisher’s exact test was used to compare categorical variables. Variables exhibiting significant differences between groups were entered into univariate and multivariate logistic regression analyses to determine factors independently predicting nosocomial bacterial pneumonia. Odds ratios (ORs) and 95% confidence intervals (CIs) were calculated using logistic regression models. We used multivariate logistic regression models to evaluate ORs, and the forward selection method was employed to assess the associated factors with a *p* value of <0.1 from univariate analyses. A *p* value of <0.05 was considered statistically significant. ICU admission proportion, MV dependence, and the duration of SARS-CoV-2 shedding were evaluated using a Kaplan–Meier curve and the log-rank test. Two-sided tests were used and considered statistically significant for *p* values less than 0.05. All statistical analyses were performed using IBM SPSS Statistics for Windows/Macintosh, Version 25.0 (IBM Corp., Armonk, NY, USA).

## 3. Results

During the study period, 296 patients with suspected severe COVID-19 infection were admitted to four ICUs. Among these patients, 224 were excluded because of the negative results of SARS-CoV-2 RT-PCR sampled at least twice. Finally, 72 patients with severe COVID-19 infection confirmed by RT-PCR were enrolled in the study, including 48 mechanically ventilated patients (Figure 1).

The patients’ demographics and clinical data are presented in Table 1. Among the enrolled patients, 30 patients (41.6%) developed HAP/VAP with positive bacterial cultures from respiratory specimens. The median (IQR) duration from symptom onset to secondary pneumonia diagnosis was 21.50 (16.50–29.25) days. In total, 141 positive cultures were obtained, including 59 of *Stenotrophomonas maltophilia* (42%), 20 of *Acinetobacter* species (14%), and 17 of *Pseudomonas aeruginosa* (12%; Figure A1). In the patients with severe COVID-19 infection who developed secondary bacterial pneumonia, higher levels of procalcitonin, lactate dehydrogenase (LDH), lactate, and D-dimer at admission were observed compared with the levels in those without bacterial pneumonia. Furthermore, a greater need for mechanical ventilator support and more use of ECMO and enoxaparin were observed in the HAP/VAP group. The patients with nosocomial pneumonia were exposed to higher accumulated dexamethasone. Moreover, there was a higher percentage of patients who received more than 60 mg of dexamethasone in the HAP/VAP group (80.0%, *p* = 0.005). Greater disease severity was observed in the nosocomial pneumonia group, with a higher SOFA score at admission. Bacteremia was checked in ten patients in the secondary pneumonia cohort. Of these, four patients also reported the same pathogens in their respiratory specimens. The origin of the bacteremia in the remaining six patients might be from a catheter-related bloodstream infection or urinary tract infection other than as a result of pneumonia.

To understand the association of SARS-CoV-2 shedding with coexisting pathogens, we analyzed the distribution of the days of SARS-CoV-2 shedding between the groups (Figure 2A). Significantly prolonged viral shedding was noted in the HAP/VAP group compared to the patients without secondary bacterial infection (median: 19.50 days vs. 15.00 days, *p* = 0.006). In the survival analysis, the patients with nosocomial pneumonia had a longer duration of SARS-CoV-2 shedding (Figure 2B).

We then explored the outcomes between the two groups. Patients with nosocomial pneumonia had poorer outcomes, including more gastrointestinal bleeding, prolonged ICU stays, prolonged viral shedding days and significantly fewer ventilator-free days at Day 28 (Table 2).

In the Kaplan–Meier analysis, patients with HAP/VAP had significantly longer ICU stays (Figure 3A). In the subgroup analysis of mechanically ventilated patients, those who developed nosocomial pneumonia had a longer duration of ventilator dependence than patients without secondary infection did (Figure 3B).

To evaluate risk factors among patients who developed secondary HAP/VAP, we performed univariate and multivariate logistic regression analyses (Table 3).

In the univariate logistic regression model, we observed that male sex, a higher serum white blood cell count, a higher LDH, a higher cumulative dose of dexamethasone (calculated from admission to the respiratory culture day in the HAP/VAP cohort and from admission to ICU discharge or death in the non-HAP/VAP cohort), and a greater SOFA score at admission were significantly associated with the development of HAP/VAP. In the multivariate analysis, only a greater SOFA score on the day of admission (OR, 1.339; 95% CI, 1.115–1.608) was significant.

## 4. Discussion

This study explored the clinical characteristics of patients with severe COVID-19 infection who developed nosocomial pneumonia and the effect of viral shedding. We observed that the greater severity of the disease at initial presentation on admission increased the risk of nosocomial bacterial pneumonia in these patients. These patients had worse outcomes, including a longer duration of ICU admission and longer mechanical ventilator dependence. A longer duration of viral shedding was also observed in the patients who developed subsequent bacterial pneumonia.

Corticosteroids have been used in various diseases, such as early severe acute respiratory distress syndrome (ARDS), and may improve clinical outcome [18]. The administration of dexamethasone (6 mg) once daily for up to 10 days was demonstrated to reduce 28-day mortality in patients with severe COVID-19 infection [19]. A multicenter randomized trial revealed that the intravenous administration of 20 mg of dexamethasone once daily for five days or 10 mg once daily for five days, or until ICU discharge, may promote longer ventilator-free days over 28 days [20]. We utilized dexamethasone doses equivalent to over 60 mg or 150 mg as the threshold, and we observed that 80% of the patients with HAP/VAP had received over 60 mg, with 50% of these patients receiving over 150 mg. The present data suggested that dexamethasone for severe COVID-19 pneumonia did not have a significant effect on the secondary infection rate [21]. Compared with the dose in our study, the median dexamethasone equivalent dose in the coinfection group was 153 mg. This dose of corticosteroid was almost the same as the dose suggested by the CoDEX trial and we also did not discover an association between this dexamethasone dosage equivalent and subsequent pyogenic pneumonia. A previous study revealed that the need for ICU admission in the first 48 h after hospitalization was a predictive factor for secondary infection [22]. In our study, multivariate regression analysis results revealed that only the SOFA score at admission was associated with secondary pyogenic pneumonia. The initial presentation of multiorgan dysfunction and immune system dysregulation caused by hyperinflammation may induce the host to be more vulnerable to the subsequent secondary infection.

The knowledge of factors associated with prolonged viral shedding is still limited. Previous studies have investigated several risk factors, including female sex, old age, and longer interval from symptom onset to hospitalization [10]. One study considered corticosteroid use as an independent factor affecting delayed viral clearance [23]. Systemic corticosteroid use might slow viral shedding by impairing cell-mediated immunity, which plays a role in viral clearance [24]. However, most studies have focused on the use or nonuse of corticosteroids. In our cohort, only five patients did not receive corticosteroid therapy, and 93% of the patients all received different dosages of corticosteroid. However, whether corticosteroid prolongs viral shedding remains controversial. Some studies did not consider the early use of corticosteroids as being associated with delayed viral clearance [25]. In our cohort, we examined the real-world data of Ct values from both nasopharyngeal and endotracheal aspirates. A previous study demonstrated that the lower possibility of the transmission of SARS-CoV-2 when Ct values were over 30 [16]. Moreover, Ct values over 30 meet the criteria for deisolation in Taiwan. We observed that the patients with COVID-19 who developed bacterial coinfection had a significantly longer duration of SARS-CoV-2 shedding. Previous evidence revealed that viruses facilitate bacterial colonization by damaging the integrity of the respiratory epithelium, reducing mucociliary function, and inducing the upregulation of different receptors required for bacterial adherence [26]. Thus, viral and bacterial coinfection developed as a result of a series of complicated mechanisms. Coinfection may result in an inflammatory reaction in the host, and the main purpose of this inflammation is to eliminate pathogens, such as viruses and bacteria. However, exaggerated inflammation may lead to tissue damage and leave the host more susceptible to other infections [27]. This exaggerated inflammation may have resulted in the dysregulation of immunity. Together, these mechanisms may partially explain the prolonged viral shedding and worse clinical outcomes observed in our study. Bhatt PJ et al. also observed that patients with severe COVID-19 and secondary bacteremia were associated with poorer prognosis [28]. It was observed that a more severe initial presentation and complicated hospital course could be associated with secondary bacterial infection.

Finally, owing to a vaccine shortage from May to October in 2021, the coverage of the second vaccine dose in Taiwan was only about 30% at the study period. Our study is a retrospective study, with limited data available. Currently, the association between vaccination and severe COVID-19 infection with secondary pneumonia is still unknown.

Our study has several limitations. First, this is a single-center, retrospective cohort study with a small sample size. We observed the longer duration of viral shedding in the secondary bacterial pneumonia group. However, the relationship between prolonged viral shedding and secondary nosocomial pneumonia is still unclear. Furthermore, a larger prospective observational study is still required. Second, although we reviewed the chest imaging of all patients, it was impossible to distinguish newly developed pulmonary infiltrates caused by SARS-CoV-2 from those caused by other pathogens. Third, our hospital’s deisolation policy varied over time; therefore, we did not standardize the process of checking Ct values from respiratory specimens. Nevertheless, this might reflect the real-world situation. In addition, we evaluated the timing for assessing viral shedding in 48 patients who received intubation during their ICU stay and the median (IQR) number of days was 3.45 (2.54–4.68).

For a better understanding of the relationship between COVID-19 and secondary infections and possible factors affecting the duration of viral shedding, a well-designed prospective clinical study is required.

## 5. Conclusions

Patients with severe COVID-19 infection who developed secondary bacterial pneumonia had worse outcomes, including longer ICU stays and fewer ventilator-free days at Day 28. Greater SOFA scores at admission increased the risk of subsequent bacterial pneumonia. Furthermore, the coinfection group had a longer duration of SARS-CoV-2 shedding.

## Figures and Tables

**Figure 1 jcm-11-06796-f001:**
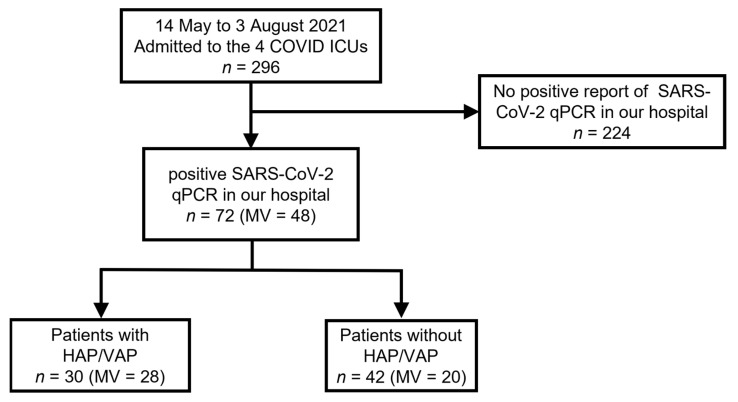
Flow chart of study. COVID: coronavirus disease; ICU: intensive care unit; SARS-CoV-2: severe acute respiratory syndrome coronavirus 2; PCR: polymerase chain reaction; HAP: hospital-acquired pneumonia; VAP: ventilator-associated pneumonia.

**Figure 2 jcm-11-06796-f002:**
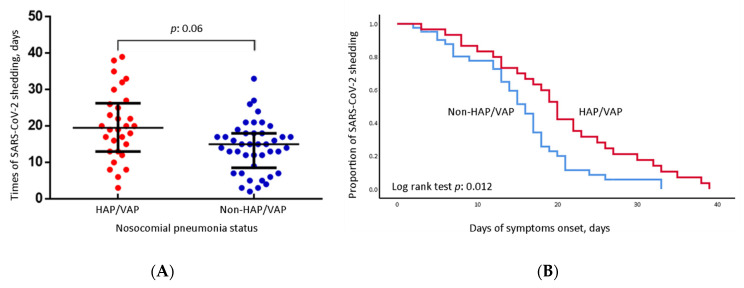
SARS-CoV-2 virus shedding and HAP/VAP. Red dots and lines: HAP/VAP group; Blue dots and lines: non-HAP/VAP group. (**A**) Distribution of SARS-CoV-2 virus shedding days categorized as HAP/VAP group and non-HAP/VAP group. We defined SARS-CoV-2 virus shedding as RNA quantification by the cycle threshold value (Ct value) being less than 30 with the open reading frame lab (ORFlab) gene as the target gene. (**B**) Patients in the HAP/VAP group had a significantly longer duration for SARS-CoV-2 virus shedding (Log rank test: *p* = 0.012).

**Figure 3 jcm-11-06796-f003:**
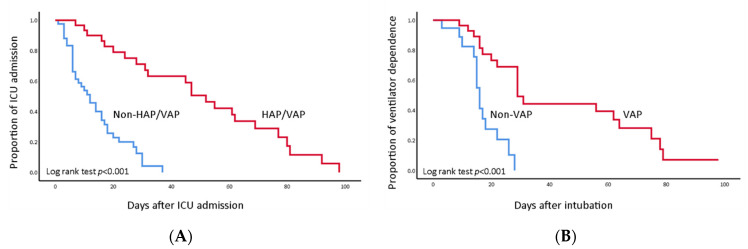
Kaplan–Meier curves for cumulative proportion of event. Red lines: HAP/VAP group; Blue lines: non-HAP/VAP group. (**A**) Patients with HAP/VAP had significantly longer times for ICU discharge (Log rank test: *p* < 0.001). (**B**) Patients with HAP/VAP had significantly longer durations of ventilator use (Log rank test: *p* < 0.001).

**Table 1 jcm-11-06796-t001:** Clinical characteristics of patients according to the HAP/VAP status.

	HAP/VAP (*n* = 30)	Non-HAP/VAP (*n* = 42)	*p*-Value
Demographics			
Age, years	67.50 (60.50–74.00)	65.00 (55.00–74.00)	0.591
Male	23 (76.7%)	24 (57.1%)	0.086
Time from symptoms onset to hospital admission, days	5.50 (2.00–9.00)	4.00 (2.00–8.25)	0.382
Time from symptoms onset to secondary pneumonia diagnosis, days	21.50 (16.50–29.25)	NA	
Underlying disease			
Cardiovascular disease	2 (6.7%)	4 (9.5%)	0.665
Diabetes mellitus	13 (43.3%)	12 (28.6%)	0.195
Cirrhosis	4 (13.3%)	7 (16.7%)	0.698
Chronic kidney disease	4 (13.3%)	6 (14.3%)	0.908
Hemodialysis	1 (3.3%)	2 (4.8%)	0.765
Malignancy	4 (13.3%)	7 (16.7%)	0.698
Laboratory tests at ICU admission			
White blood cells, 10^9^/L	6413.00 (5050.00–12,375.00)	5700.00 (3400.00–8975.00)	0.089
Lymphocytes, 10^9^/L	689.70 (427.70–877.75)	607.50 (442.13–925.05)	0.918
Albumin, g/dL	3.30 (3.10–3.62)	3.55 (3.30–4.08)	0.057
C-reactive protein, mg/dL	8.03 (3.42–14.33)	8.45 (3.00–10.03)	0.668
Procalcitonin, ng/mL	0.18 (0.11–0.82)	0.13 (0.07–0.33)	0.028
LDH, U/L	522.50 (398.00–679.00)	361.50 (283.75–555.00)	0.014
Lactate, mg/dL	18.40 (14.33–28.33)	15.20 (10.30–18.88)	0.022
D-dimer, ug/mL	1.45 (0.63–6.57)	0.80 (0.42–1.40)	0.028
Fibrinogen, mg/dL	410.05 (271.15–523.50)	440.20 (369.30–537.50)	0.395
Treatment			
Cumulative dose of dexamethasone ^a^	153.00 (66.00–186.50)	60.00 (22.50–159.25)	0.011
Dexamethasone dose > 60 mg	24 (80.0%)	20 (47.6%)	0.005
Dexamethasone dose > 150 mg	15 (50.0%)	12 (28.6%)	0.064
Tocilizumab	20 (66.7%)	20 (47.6%)	0.109
Remdesivir	24 (80.0%)	30 (71.4%)	0.408
Enoxaparin	27 (90.0%)	16 (38.1%)	<0.001
Severity scores			
SOFA at admission	7.50 (5.00–10.00)	4.00 (1.00–8.00)	0.002

Data are presented as the median (IQR) and number (%) unless otherwise indicated. ^a^ Dosage of dexamethasone equivalents were calculated from admission to the first day of bacterial culture for respiratory specimens in the HAP/VAP group and from admission to ICU discharge or death in the non-HAP/VAP group. NA: not applicable; ICU: intensive care unit; LDH: lactate dehydrogenase; SOFA: sequential organ failure assessment.

**Table 2 jcm-11-06796-t002:** Clinical outcomes of patients according to HAP/VAP status.

	HAP/VAP (*n* = 30)	Non-HAP/VAP (*n* = 42)	*p*-Value
**Clinical course and outcomes**			
Mechanical ventilator	28 (93.3%)	20 (47.6%)	<0.001
ECMO	5 (16.7%)	0 (0.0%)	0.006
Prone position	9 (30.0%)	8 (19.0%)	0.281
Renal replacement therapy	2 (6.7%)	4 (9.5%)	0.665
GI bleeding events	14 (46.7%)	4 (9.5%)	<0.001
SARS-CoV-2 shedding days	19.50 (13.00–26.25)	15.00(8.50–18.00)	0.006
ICU stay, days	39.50 (17.00–63.75)	10.00 (6.00–18.00)	<0.001
Ventilator free days at Day 28	0.00 (0.00–0.00)	10.50 (0.00–13.00)	0.006
ICU mortality	7 (23.3%)	6 (14.3%)	0.325
In-hospital mortality	7 (23.3%)	7 (16.7%)	0.481

Data are presented as the median (IQR) and number (%) unless otherwise indicated. ECMO: extracorporeal membrane oxygenation; GI: gastrointestinal; ICU: intensive care unit.

**Table 3 jcm-11-06796-t003:** Association of HAP/VAP by univariate and multivariate logistic regression analyses.

	Univariate ^a^			Multivariate ^a^		
Variables	OR	95% CI	*p*-Value	OR	95% CI	*p*-Value
Male	0.406	0.143–1.152	0.090			
White blood cell, 109/L	1.000	1.000–1.000	0.094			
Albumin, g/dL	0.857	0.429–1.711	0.662			
Procalcitonin, ng/mL	1.199	0.844–1.703	0.311			
LDH, U/L	1.003	1.000–1.005	0.029			
Lactate, mg/dL	1.004	0.984–1.024	0.720			
D-dimer, ug/mL	1.038	0.984–1.094	0.168			
Cumulative dose of dexamethasone ^b^	1.007	1.001–1.013	0.017			
SOFA at admission	1.239	1.069–1.436	0.004	1.339	1.115–1.608	0.002

OR: odds ratio; CI: confidence interval; LDH: lactate dehydrogenase; SOFA: sequential organ failure assessment. ^a^ Odds ratios and 95% confidence interval were derived from logistic regression analysis. ^b^ Dosage of dexamethasone was calculated from admission to the first day of bacterial culture from respiratory specimens in the nosocomial pneumonia group and from admission to ICU discharge or death in the non-HAP/VAP group.

## Data Availability

The datasets used or analyzed during the current study are available from the corresponding author on reasonable request.

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
