# Peer review of "Clinical Outcomes and Prolonged SARS-CoV-2 Viral Shedding in ICU Patients with Severe COVID-19 Infection and Nosocomial Bacterial Pneumonia: A Retrospective Cohort Study"

_jcm, 2022, doi:10.3390/jcm11226796_

Round 1
Reviewer 1 Report
A well designed and well conducted study.
Both in the description of the adopted methods and in the presentation of the results: no opaque point. The discussion of the results is detailed and correct; the limitations of the research are honestly made explicit in detail ("a single center, restrospective cohort study"; difficulties in the distinction. at the chest imaging. between the lung infiltrates caused by SARS-CoV-2 and the ones caused by other pathogens; troubles from the time variations of the adopted criteria for deisolation, determining a lack of standardization of the Ct values control from respoiratory specimens). I agree with the Authors' final statements: - the pattern designed by their results reasonably depicts "a real world situation"; - downstream of their results, a prospective clinical study about the relationships bewteen COVID-19, secondary infections during hospitalization and duration of the viral shedding is not only justified, bur required.
If the authors agree, in my opinion useful to expand the discussion about the possibility of infectiousness of patient with a CT between 30 è 39: but this is not vital, for the authors evidenced that 30 is simply cut-off value adopted by their hospital for the dismission from isolation (under the absumption of no infectiousness in the presence of Ct values of 30 or more).
Author Response
We appreciated your rigorous review and suggestions. We further emphasized the association between Ct value and infectiousness in discussion. A previous study demonstrated the lower possibility of transmission of SARS-CoV-2 when Ct values were over 30. Moreover, Ct values over 30 meet the criteria for deisolation in Taiwan. We have revised our manuscript. Please see line 251 in the revised manuscript. In addition, the reference we cited showed that no virus was isolated when Ct value was over 30.(1)
- Young BE, Ong SWX, Ng LFP, Anderson DE, Chia WN, Chia PY, et al. Viral Dynamics and Immune Correlates of Coronavirus Disease 2019 (COVID-19) Severity. Clin Infect Dis. 2021;73(9):e2932-e42.
Reviewer 2 Report
This study indicates that patients with severe COVID-19 who develop VAP/HAP have increased duration of SARS-CoV2 shedding. This conclusion is inaccurate since most VAP/HAP developed in the later part of hospitalization (authors indicate 21.5 days) by which time many patients probably had cleared SARS-CoV2 (median duration of viral shedding 19.5 days). Therefore, one cannot imply that prolonged viral shedding was related to VAP/HAP.
It appears that those who went on to develop VAP/HAP were already critically ill and so were at risk for nosocomial infections. The association between VAP/HAP and viral shedding noted in this study is probably due to both these conditions being common in critically ill patients. Therefore, the severity of the primary SARS-CoV2 infection (as indicated by high LDH/procalcitonin/d dimer/SOFA scores at admission, longer ICU and ventilator days) likely caused the prolonged shedding and not VAP/HAP. Several previous studies have shown that greater disease severity is associated with prolonged viral shedding.
Author Response
We totally agreed and appreciated your insightful input. Recent research demonstrated that patients with severe COVID-19 infection developing secondary bacterial infection might be associated with more severe initial presentation (1). As for prolongation of SARS-CoV-2 viral shedding, a retrospective study revealed that the median duration of viral shedding was significantly longer in patients with severe COVID-19 infection than in those with mild disease (2). In our study, higher SOFA score at admission was associated with secondary bacterial pneumonia in multivariate regression analysis. However, the relationship between prolonged duration of SARS-CoV-2 shedding and secondary bacterial infection is still unclear. We added this point in the limitation of discussion: ‘’ We observed longer duration of viral shedding in the secondary bacterial pneumonia group. However, the relationship between prolonged viral shedding and secondary nosocomial pneumonia is still unclear. Further larger prospective observational study is still required.” Please read line 270 in the revised manuscript.
- Bhatt PJ, Shiau S, Brunetti L, Xie Y, Solanki K, Khalid S, et al. Risk Factors and Outcomes of Hospitalized Patients With Severe Coronavirus Disease 2019 (COVID-19) and Secondary Bloodstream Infections: A Multicenter Case-Control Study. Clin Infect Dis. 2021;72(12):e995-e1003.
- Zheng S, Fan J, Yu F, Feng B, Lou B, Zou Q, et al. Viral load dynamics and disease severity in patients infected with SARS-CoV-2 in Zhejiang province, China, January-March 2020: retrospective cohort study. BMJ. 2020;369:m1443.
Reviewer 3 Report
The authors presented a retrospective study on patients with severe COVID-19 infection. They concluded that patients with severe COVID-19 infections with nosocomial pneumonia had prolonged and more complicated outcomes.
It is very hard to read the manuscript, including all the corrections in the text.
The authors performed the RT-PCR measurement on the patient's samples, but in the method section, there is no information about the control, which was included in the measurement to set up the lowest Ct value.
Author Response
- We appreciated your valuable comments. English editing was performed for this revised manuscript for better reading and understanding of the manuscript.
- We requested the of values of limit of detection (LoD) from the Roche Diagnostics. The limit of detection (LoD) determines the lowest detectable level of SARS-CoV-2 at which greater or equal to 95% of all replicates test positive. The LoD was determined via a cultured and isolated virus from a US patient (USA-WA1/2020, catalog number NR-52281, lot number 70033175, 2.8E+05 Median tissue culture infectious dose (TCID)50/mL). The concentration level with observed hit rates over or equal to 95% was 0.009 TCID50/mL for SARS-CoV-2 and the Probit-predicted 95% hit rate was 0.007 TCID50 /mL (95% CI: 0.005-0.036). The concentration level in a dilution series with observed hit rates over or equal to 95% was 46 copies/mL for SARS-CoV-2. The Probit model 95% LoD was about 25 copies/mL (95% CI: 17-58 copies/mL). We have revised the manuscript. Please read line 113 in the revised manuscript.
Round 2
Reviewer 2 Report
I agree with the revisions. Authors could consider stating that poor outcomes in HAP/VAP cohort could be due to the bacterial co-infection or alternatively these patients had complicated hospital course leading to acquiring risk factors for developing co-infection. This is stated in the study referenced by the authors : " Risk Factors and Outcomes of Hospitalized Patients With Severe Coronavirus Disease 2019 (COVID-19) and Secondary Bloodstream Infections: A Multicenter Case-Control Study."
Author Response
We appreciated your valuable comments. We have added the statement in the discussion: “Bhatt PJ et al also observed that patients with severe COVID-19 and secondary bacteremia were associated with poorer prognosis. It was observed that a more severe initial presentation and complicated hospital course could be associated with secondary bacterial infection.” Please read line 265 in the revised manuscript. We also cited the suggested reference in our revised manuscript. Please read line 405. In addition, the English editing service has been performed by the JCM, English editing ID: english-53481.